# Metallic Nanosystems in the Development of Antimicrobial Strategies with High Antimicrobial Activity and High Biocompatibility

**DOI:** 10.3390/ijms24032104

**Published:** 2023-01-20

**Authors:** Karol Skłodowski, Sylwia Joanna Chmielewska-Deptuła, Ewelina Piktel, Przemysław Wolak, Tomasz Wollny, Robert Bucki

**Affiliations:** 1Department of Medical Microbiology and Nanobiomedical Engineering, Medical University of Bialystok, 15-222 Bialystok, Poland; 2Independent Laboratory of Nanomedicine, Medical University of Bialystok, 15-222 Bialystok, Poland; 3Institute of Medical Science, Collegium Medicum, Jan Kochanowski University of Kielce, IX Wieków Kielce 19A, 25-317 Kielce, Poland; 4Holy Cross Oncology Center of Kielce, Artwińskiego 3, 25-734 Kielce, Poland

**Keywords:** antibiotic resistance, metallic nanoparticles, synergy, biocompatibility, modulation of antimicrobial activity

## Abstract

Antimicrobial resistance is a major and growing global problem and new approaches to combat infections caused by antibiotic resistant bacterial strains are needed. In recent years, increasing attention has been paid to nanomedicine, which has great potential in the development of controlled systems for delivering drugs to specific sites and targeting specific cells, such as pathogenic microbes. There is continued interest in metallic nanoparticles and nanosystems based on metallic nanoparticles containing antimicrobial agents attached to their surface (core shell nanosystems), which offer unique properties, such as the ability to overcome microbial resistance, enhancing antimicrobial activity against both planktonic and biofilm embedded microorganisms, reducing cell toxicity and the possibility of reducing the dosage of antimicrobials. The current review presents the synergistic interactions within metallic nanoparticles by functionalizing their surface with appropriate agents, defining the core structure of metallic nanoparticles and their use in combination therapy to fight infections. Various approaches to modulate the biocompatibility of metallic nanoparticles to control their toxicity in future medical applications are also discussed, as well as their ability to induce resistance and their effects on the host microbiome.

## 1. Introduction

Antimicrobial resistance (AMR) is a natural phenomenon that occurs when microorganisms are exposed to antimicrobial agents [1]. The speed of this natural process has been drastically affected by the use of antibiotics not only in medicine, but also in other sectors. The list of causes of increasing antibiotic resistance includes: (i) excessive use of antimicrobials in veterinary medicine/agriculture, where the addition of antibiotics to feed for farm animals not only ensures the prevention of intestinal infections but also results in improved absorption of nutrients and, thus, faster weight gain while causing the selection and development of bacterial strains with mechanisms of resistance and further release of these strains into the environment [2], (ii) over-use of antimicrobials due to the over-prescription of antibiotics (approximately 90% of all antibiotic prescriptions are issued by general practitioners and respiratory infections are the main reason for prescribing them, however, they are mainly caused by viruses) [3], (iii) improper selection of doses, which prevents the complete elimination of pathogens, which then favors changes in gene expression, increased mutagenesis or horizontal gene transfer and, thus, increasing resistance to antibiotics and the spread of these strains, especially in the hospital environment [4,5], (iv) the duration of antibiotic therapy, with a longer duration of therapy being associated with an increased risk of antimicrobial resistance [6]. Due to the overuse of antimicrobials in veterinary medicine [7], inappropriate use of antibiotics resulting from a lack of knowledge on the principles of rational antibiotic therapy [3], biofilm formation by pathogenic microorganisms [8], an increasing number of infections with multidrug-resistant strains (MDR) has been observed [9], making antimicrobial resistance one of the biggest public health challenges of our time [10,11]. A report published by the Centers for Disease Control (CDC) indicates the highest number of infections in United States caused by drug-resistant *Streptococcus pneumoniae* with 900,000 cases, followed by 550,000 infections caused by drug-resistant *Neisseria gonorrhoeae*, and in third place were 448,400 infections caused by drug-resistant *Campylobacter* [12]. Importantly, it is estimated by World Health Organization (WHO) that the drug resistance of microorganisms is already responsible for at least 700,000 deaths each year, including 230,000 people dying from multidrug-resistant tuberculosis [13]. The seriousness of the problem in the treatment of infectious diseases is evidenced by analyses conducted by scientists, who predict that by the end of 2050, untreatable infections will be the most common cause of death, causing more than 10 million deaths per year, overtaking cancer and cardiovascular diseases [14].

The problem of increasing resistance is a growing concern as the number of new antibiotics approved since the late 1970s has declined [15]. Evidently, the propensity of microbes to develop resistance occurs much faster than the ability of humans to develop new agents; therefore, new antimicrobial compounds are being sought and nanomaterials appear to be a promising alternative to conventional antimicrobials due to their unique physical and chemical properties [16,17].

Nanotechnology is the science of materials/devices defined by size (the nanoscale range is 1–100 nm in one dimension). The term nanotechnology was introduced by American physicist and Nobel Prize winner Richard Feynman in 1959 during a lecture entitled “There’s Plenty of Room at the Bottom”. Dr. Richard Feynman considered some of the consequences of the possibility of manipulating matter on the atomic scale and mentioned the ability to create nanoscale machines [18]. Fifteen years later, the term “nanotechnology” was defined by Professor Norio Taniguchi from Tokyo Science University in the 1974 paper: ‘Nano-technology’ mainly consists of the processing of, separation, consolidation, and deformation of materials by one atom or by one molecule” [19].

A steadily increasing number of reports indicate that nanomaterials may be also developed as alternative to currently used antibiotics and antifungal drugs [20]. Nanomaterials, among which four categories can be distinguished, depend on their material type. (i) The first group includes carbon-based nanomaterials such as fullerenes, carbon nanotubes, graphene and its derivatives, graphene oxide, nanodiamonds, or carbon-based quantum dots [21]. In their pure state, most carbon-based nanomaterials have limited antimicrobial capacity and show low selective toxicity against bacteria over mammalian cells. By modifying their physicochemical properties, their antimicrobial activity and targeting efficiency can be modulated [22] through surface functionalization to modulate physicochemical parameters or modification of their synthesis methods [23] (using covalent and non-covalent modification, among others [24]). In addition, to improve the water solubility and dispersion of carbon-based nanomaterials, surfactants and polymer are used to increase both the probability of contact and the strength of interaction with bacteria [25]. In order to enhance antibacterial efficacy, carbon-based nanomaterials are functionalized with functional groups and bioactive molecules [26]. The second group represents (ii) inorganic-based nanomaterials consisting of metal (e.g., Au, Ag, Pt) [27], metal oxide NPs (nanoparticles) (e.g., TiO_2_, MnO, ZnO) and semiconductors such as silicon and ceramics [28]. Inorganic-based nanomaterials are of great interest due to a number of features such as optical properties including surface plasmon resonance (SPR) with the ability to control optical field, the possibility to modify the surface of nanoparticles to control solubility, stability and interaction with the environment (it is possible, among other things, to increase the circulation time of NPs by reducing non-specific uptake by the mononuclear phagocyte system), mechanisms of action quite different from those described for traditional antibiotics, irrespective of the pathogen resistance mechanism, synthetic versatility, which allows the control of their size, shape and surface properties, surface functionalization of NPs with an appropriate functional groups for the labelling, targeting and conjugation of pharmacological molecules, synthesis by simple, cost-effective, and easy methods [29,30,31,32,33,34]. Another negative trait of inorganic-based nanomaterials is their toxicity, which can be modulated by changing the shape and size of the particles and modifying their surface, leading to nanoparticles with desired properties but without toxic effects [35,36]. The third group represents (iii) organic-based nanomaterials which include molecules made of organic material as cationic polymers NPs, solid lipid NPs, lipid NPs, biomimetic NPs, dendrimer nanoparticles or protein-based NPs [37]. A key advantage of organic-based nanomaterials NPs is the tunability of the lipid layer, which can be further functionalized to produce nanomaterials with the desired properties. In addition, they have advantageous characteristics such as chemical diversity, high loading capacity and intrinsic biodegradability [38] and biocompatibility [39,40]. However, compared to inorganic materials, they are less stable by nature, especially at higher temperatures [41], and the presence of potential problems related to immunogenicity and challenges in loading of a wide variety of drugs [42] or poor mechanical and processing properties or insolubility in common organic solvents [43]. The last group represents (iv) composite-based nanomaterials that are comprised of two or more components at the nanoscale where mutual contact interfaces occur between the individual components. Composites can be any combination of metal, carbon, or organic based-NMs (nanomaterials) with any form of metal or polymer materials [44]. The advantages of composite-based nanomaterials include the film uniformity, biocompatibility, available hydroxyl and carboxyl groups or amines, improve physical properties of ions and their releasing, possibility of functionalizing the surface, environmental stability, simple doping process or tunable conductivity [45,46]. The disadvantages of these nanomaterials include uncertain cytotoxicity, component stability, long-term stability, structural integrity, mechanical and corrosion properties or the tendency of nanomaterials to agglomerate [47,48]. From among the above-mentioned categories, the most promising are metallic nanoparticles, which show strong antimicrobial activity both against planktonic bacteria and in biofilm form in a large number of studies, which is why this review focuses on metallic NPs.

Due to their unique physicochemical properties such as (i) a large surface to the volume ratio, (ii) the ability to functionalize with diagnostic and therapeutic factors, (iii) ease of modification of the method of synthesis, (iv) antibacterial and immunomodulatory properties [49,50,51,52,53,54,55], nanoparticles are of growing interest in medicine. It is noteworthy that due to the nanometer scale size and appropriate surface charge, a strong interaction of nanoparticles with the biological membranes of the pathogen is possible [56,57]. Moreover, in respect of their relatively low potential to induce drug resistance [58,59], metal nanoparticles are proposed as an alternative to antimicrobial agents. They are also receiving increasing recognition as highly effective drug carriers [60].

The mechanism of action of metallic nanoparticles includes, among others: (i) disruption of the cell walls, thus, increasing their permeability as a result of electrostatic interaction between negatively charged molecules of the cell wall of the microorganism and positively charged nanoparticles resulting in a leakage of cytoplasmic contents [61,62] and causing membrane potential disorder [63]; (ii) another mechanism comprises of the generation of toxic Reactive Oxygen Species (ROS). Oxidative stress leads to oxidation of glutathione, disrupting the antioxidant defense mechanisms of bacteria against ROS. The excessive production of ROS causes disturbances in redox homeostasis, which results in oxidative stress, thus, affecting the membrane lipids and modifies DNA as well as the protein structure [61,64]; (iii) a further mechanism involves the binding to intracellular components among other things, causing damaged DNA, proteins and inhibition of the enzymatic activity [65]. The interaction of metallic nanoparticles with DNA can denature or shear the DNA and disrupt cell division [66,67]. In addition, metallic NPs can inhibit protein synthesis by denaturing ribosomes [68]. As a result of the additive effect of the above factors, (iv) apoptotic cell death eventually occurs [69]. A schematic representation of the different mechanisms of nanoparticles action is illustrated in Figure 1.

Here, we provide a throughfall characterization and discussion of the latest achievements in synthesis and design of metallic nanoparticles and metallic nanoparticle-based nanosystems as potent antimicrobials with the potential to be used for the treatment of drug-resistant bacterial and fungal infections.

## 2. Synergistic Effects of Metallic Nanoparticles

Over the past few decades, antibiotic-resistant bacteria have become increasingly prevalent; the number of infections caused by multidrug-resistant (MDR) bacteria is increasing and the risk of untreatable infections is rising [70].

Among metallic nanoparticles, silver (Ag), gold (Au), copper oxide (CuO), iron oxide (Fe_3_O_4_) titanium oxide (TiO_2_) or zinc oxide (ZnO) are commonly used as antimicrobial agents after their strong antimicrobial activity is well known [16,71]. There are many studies showing that various metal and metal oxide nanoparticles exhibit biocidal activity against gram-positive and gram-negative bacteria, fungi or viruses [72]. A key influence on the antimicrobial properties of metallic NPs is their high specific surface area high surface-to-volume ratio and nanoscale size, which allows strong interaction with the membranes of micro-organisms causing its disruption, penetration into cells followed by damage to internal cellular structures ultimately leading to the cell death [52]. The mechanisms associated with metallic nanoparticles ability to overcome antibiotic resistance involved their unique physicochemical properties enabling the exploitation of multiple novel bactericidal pathways to achieve antimicrobial activity [73]. Due to the binding between metal ions and microbials’ biomolecules, which is generally non-specific, metallic nanoparticles exhibit a broad spectrum of activity [74]. Specific metal ions such as iron, zinc or copper are essential for the biochemistry of life in all organisms, and their deficiency can cause damage to the structure of cell membranes and DNA or disrupt enzymatic functions [75]. However, an excess of these ions or the presence of other, less essential ions such as gold or silver can be lethal to pathogens’ cells. Released from the extracellular space, metal ions are able to enter the cell and disrupt biological processes where, inside the cell, they can induce the production of ROS and affect cellular structures by disrupting cellular functions as a result of forming strong coordination bonds with nitrogen, oxygen and sulfur atoms, which are abundant in organic compounds and biomolecules [61].

Due to the increasing prevalence of microbial resistance, combinations of nanoparticles and antimicrobials have been shown to possess superior efficacy compared to antimicrobials alone [76,77,78]. Such combinations can reduce the development of antimicrobial resistance as well as shorten the duration and dose requirements of antimicrobial treatment [79,80]. The use of combination therapy is common in clinical practice for many reasons, including: (i) the prevention of antimicrobial resistance [81], (ii) antimicrobials can mutually enhance antimicrobial activity [77], (iii) when a critically ill patient is admitted with suspected sepsis of unknown etiology, several antimicrobials are used to broaden the spectrum against unknown pathogenic species [82,83], (iv) killing bacteria in a dormant state [84].

Based on the type of components that comprise the combination of metallic nanoparticles, they can be divided into several categories: (i) monometallic nanoparticles, (ii) metallic nanoparticles in combination with conventional antibiotics/fungicides or compounds other than antimicrobial agents, (iii) multimetallic nanoparticles alone and (iv) in combination with antibiotics/fungicides, (v) metallic nanoparticles, whose surface has been further functionalized with antibiotics/fungicides or compounds other than antimicrobial agents. The functionalization of metallic NPs surfaces with the desired compound utilizes various types of covalent and non-covalent bonds—these include electrostatic forces, hydrogen bonds and van der Waals interactions, resulting in the integration of a variety of organic and inorganic molecules at the nanoscale [85]. In order to form covalent and non-covalent bonds between ligands and NPs surfaces, a number of linker molecules are used, such as organic materials, within which various polymers (polyethylene glycol (PEG), polyvinyl alcohol (PVA), chitosan, dextran, alginate, polyacrylic acid, citrates, phosphates, amines [86]) or inorganic substances (metals and metal oxides, silicas [87,88,89]) are used. The non-covalent functionalization approach is based on a large number of weak interactions such as ionic interactions, van der Walls, hydrophobic interactions, electrostatic interactions, hydrogen bonds that are applied to metallic and silica nanoparticles [90,91,92]. The advantages of using non-covalent modifications include the simplicity and lack of influence on the structures of the particles used and their interaction with docked biological substances, while the disadvantage is that non-covalent interactions are easily influenced by factors such as pH or ionic strength [93]. The surface modification of NPs using covalent bonds can be achieved using a number of alternative approaches, depending on the composition of the NPs [94,95,96] by means of modifications at several levels using sequential functionalization, so that structures with multiple functions can be obtained [97,98].

### 2.1. Monometallic Nanoparticles

Monometallic NPs consist of a single metal species, which, depending on the atomic type and properties, may exist in various forms such as metallic, magnetic, transition metal and oxide. Monometallic NPs are the most popular inorganic nanoparticles, which represent a promising solution in the fight against resistance to traditional antibiotics, not only because of their completely different mechanisms of action from commonly used antibiotics, showing activity against bacteria that have developed resistance, but also because they target many biomolecules that impede the development of resistant strains [99].

Among monometallic NPs, silver and gold nanoparticles are leading the way. Silver NPs are of great interest as antimicrobial agents due to their exceptional antimicrobial activity against a broad spectrum of pathogenic microorganisms [100,101]. Within the antimicrobial action of silver nanoparticles, three main mechanisms of action can be distinguished: firstly, the interaction and penetration of nanoparticles into the membrane of microorganisms [102], which results in protein inactivation and membrane lipid peroxidation, leading to structural modification of membrane integrity, transport protein dysfunction and leakage of cellular contents [103,104]. Secondly, there is damage to intrinsic structures, which triggers ROS generation, leading to the disruption of redox hemostasis, affecting the Na^+^/K^+^ ATPase pump and signal transduction pathways [105]. As a result of the interaction of ions and nanoparticles with DNA, protein inactivation occurs, ultimately leading to cell death [106]. Thirdly, there is the release of Ag^+^ ions (whose rate of release depends largely on the size, shape, concentration, capping agent or colloidal state of NPs [107,108]), which occurs in parallel with the other two, which, due to their size and charge, can interact with cell components to alter metabolic pathways and even genetic material [109,110]. It is also important to keep in mind the type of bacterial species that respond differently to the activity of Ag NPs, which is caused by the different composition and thickness of the cell wall [111].

Gold nanoparticles (AuNPs) are one of the most important nanoparticles due to their simple and controlled synthesis, inertness, biocompatibility and low toxicity compared with other nanomaterials. Gold nanoparticles, such as silver nanoparticles, disrupt the integrity and structure of the cell membrane, causing leakage of intracellular components [112,113,114]. It can be compared to apoptosis-like cell death, where gold nanoparticles cause depolarization of the bacterial cell membrane and a continuous increase in the concentration of calcium ions in the cytoplasm, induction of DNA fragmentation, resulting in apoptosis-like death (overexpression of caspase-subunit proteins was observed as well) [115]. Additionally, membrane potential is altered and ATP synthase activity is reduced, resulting in metabolic dysfunction [116]. On the other hand, our studies with gold nanoparticles coated with ceragenin CSA-131 confirmed that cell membrane depolarization and cytoplasmic protein leakage occur when ESCAPE strains are targeted [117] (Figure 2 adopted from [117]).

The interaction of Au NPs with intracellular biomolecules results in translation inhibition [118]. The antimicrobial mechanism of Au NPs also involves an increase in intracellular ROS levels [119]. Additionally, our previous results show that gold nanoparticles display antibiofilm activity against *Candida* by reducing pathogen cell adhesion, resulting in the inhibition of biofilm growth. Interestingly, peanut shaped gold nanoparticles were found to reduce the viscosity of the biofilm formed by *Pseudomonas*, which may be important in the case of cystic fibrosis where thick mucus are formed, making it difficult for antimicrobial agents to penetrate and subsequently eradicate the pathogens causing infection. (Figure 3 and Figure 4 from [120,121] respectively).

Among other factors, antimicrobial activity is strongly influenced by the shape [107] and size [122] of nanoparticles, even for nanoparticles with the same surface-to-volume ratio. Typically, nanoparticles of smaller size have higher antimicrobial activity [123,124], but there are reports that larger nanoparticles are more effective, which may suggest that size alone is not the most important factor in their activity and toxicity [125] and, thus, it can be hypothesized that with certain metallic NP systems, antimicrobial activity may be largely controlled by the extent of electrostatic interactions with the microbial cell wall. In order to verify this hypothesis indicating shape influence on the activity of metallic nanoparticles, metallic NPs in different shapes were synthesized, then their antimicrobial activity was evaluated. Cheon et al. [126] synthesized Ag NPs with spherical, triangular plate and disk shapes in aqueous solution. Based on the zone of bacterial growth inhibition, the highest antibacterial activity was recorded for spherical Ag NPs, followed by disc shaped Ag NPs, while the lowest activity was recorded for triangular plate Ag NPs. The difference in antimicrobial activity of these Ag NPs was explained by the release rate of Ag ions from the surface. In another study by El-Zahry et al. [127] spherical, triangular and hexagonal Ag NPs of the same size were synthetized by chemical reduction. The results of this work show that hexagonal Ag NPs exhibit the highest antimicrobial activity compared to spherical and triangular NPs, which is associated with a larger surface area, allowing a stronger antimicrobial effect. In our research with metallic nanoparticles, we also observed shape-dependent activity of Au NPs. For ceragenin-containing gold nanoparticles in the shape of rods (AuR NPs@CSA-131), peanuts (AuP NPs@CSA-131) and stars (AuS NPs@CSA-131), the antimicrobial activity of peanut-shaped gold nanoparticles was lower compared to those in rod and star shapes [117]. Another important factor governing the antimicrobial activity of NPs is their charge. Positively charged metallic nanoparticles are able to alter the function of the electron transport chain in bacteria, leading to the neutralization of the surface electrical charge of the bacterial membrane and altering its permeability, ultimately causing bacterial death [128]. However, there are some discrepancies regarding the activity of metallic nanoparticles against gram-positive and gram-negative bacteria. Some researchers reported that gram-positive bacteria are more sensitive to nanoparticles, due to the fact that the cell wall structure of gram-negative bacteria is more complex [129]. On the opposite side, researchers believe that gram-negative bacteria are more susceptible to antibacterial Ag nanoparticles due to the easier passage of Ag ions through the thinner cell walls [130].

The activity of metallic nanoparticles can be modulated by doping with suitable compounds, such as transition metals. Singh et al. evaluated how the antimicrobial activity of ZnO NPs doped with Fe, and CdS doped with Fe and Co would change. The results showed that Fe-doped ZnO nanoparticles exhibited decreased antibacterial activity against gram-negative bacteria, which could be due to the decrease in the positive surface charge carried by the nanoparticles, and also the change in the surface morphology of ZnO nanoparticles as a result of Fe doping, while Fe atom-doped CdS NPs increased the antibacterial activity of the nanoparticles with no change in the activity of cobalt-doped CdS NPs [131]. As a result of the doping of metallic nanoparticles with transition metals, changes in the NPs’ charge and size were observed, which is a key factor in their antimicrobial activity. An increase in the antimicrobial activity of copper ferrite NPs doped with nickel [132] or Mg-doped ZnO nanoparticles [133] was also observed. In one of our studies, magnetic nanoparticles functionalized with gold displayed strong bacteriostatic activity against *Pseudomonas aeruginosa*. We assumed that the gold present on the surface of magnetic nanoparticles interacts with bacterial proteins through disulfide bonds, which can have a significant impact on the microbial cells metabolism and redox system of [134].

The antimicrobial properties of metallic nanoparticles can be improved by increasing the solubility of nanoparticles in aqueous media. By synthesizing silver nanoparticles using an aqueous solution of an extract from the plant *Pulicaria glutinosa*, it was determined how the solubility of Ag NPs affects their activity. The results indicate that with the increase in the solubility of silver nanoparticles obtained by increasing the concentration of the plant extract used for the synthesis (from 4% to 21%), a decrease in the values of the half maximal inhibitory concentration were observed against *Escherichia coli*, *Pseudomonas aeruginosa*, *Staphylococcus aureus* and *Micrococcus luteus* strains [135]. Although metallic nanoparticles exhibit high antimicrobial activity to effectively combat pathogens, their inherent characteristic of low solubility causes a significant loss of antimicrobial capacity and leads to increased toxicity [136], where solubility determines many important properties of nanoparticles, including their surface area, which makes it possible to control the interaction between nanoparticles and microorganisms [136].

The activity of metallic nanoparticles can also be governed by modulating their surface to improve their functionality as antimicrobial compounds. By functionalizing the surface of iron oxide nanoparticles with L-tyrosine, a significant difference in antimicrobial activity was observed between nanoparticles whose surface was modified, compared with non-functionalized iron oxide nanoparticles [137]. For the non-functionalized iron oxide nanoparticles, no antimicrobial activity was observed against *Staphylococcus aureus* and *Salmonella typhimurium*, whereas the functionalized iron oxide nanoparticles showed antimicrobial activity against these strains, as a result of the formation by L-tyrosine functionalization of more stable NPs with different functional groups on the surface, providing a better binding interaction with microorganisms [138]. Nijonshuti et al. [139] compared the activity of Ag NPs and Ag NPs whose surface was functionalized with polydopamine (PDA). The results indicated that the PDA coating acted in synergy with Ag NPs, significantly increasing the potency of Ag NPs against bacteria, and suggest that higher valence/oxidation state increases the antimicrobial potency of Ag [140] and coordination between Ag and PDA mainly through the catechol group, which may play an important role in regulating the antimicrobial activity of PDA-Ag NPs [141]. In our study, as a result of the functionalization of the surface of gold nanoparticles with a cationic steroid antimicrobial (CSA), we obtained enhanced antimicrobial activity against gram-positive bacteria, gram-negative bacteria, and fungi, regardless of the resistance mechanism, as well as against microorganisms both in planktonic form and growing in biofilm as a result of the permeabilization of the cell membrane and release of protein content and generation of ROS [117,119,142]. Similarly, in the case of magnetic nanoparticles that were functionalized with compounds such as PBP10 peptide, 1,4-dihydropyridine, ceragenin, LL-37, chlorhexidine, increased antimicrobial activity of the functionalized nanoparticles was obtained compared with nanoparticles alone and compounds in the free form [143,144,145,146,147].

The antimicrobial activity of metallic nanoparticles is influenced by their method of synthesis. In a study by Garibo et al., silver nanoparticles synthesized by green synthesis using an extract from *Lysiloma acapulcensis* possessed higher antimicrobial potency than chemically produced Ag NPs. Antimicrobial activity was determined using the disk diffusion method and minimal inhibitory concentrations (MICs) and minimal bactericidal concentrations (MBCs) against four clinical strains: *Escherichia coli*, *Pseudomonas aeruginosa*, *Staphylococcus aureus* and *Candida albicans*. Both a larger zone of inhibition and lower MICs and MBCs of biogenic Ag NPs were observed in comparison with Ag NPs synthesized by the chemical method [148]. In the case of the study provided by Ghetas et al. the antimicrobial activity of biologically synthesized Ag NPs using an extract from *Origanum vulgare* and Ag NPs synthesized chemically was assessed. Using the disk-diffusion method, their activity against *Streptococcus agalactiae*, *Aeromonas hydrophila*, *Vibrio alginolyticus*, *Aspergillus flavus*, *Fusarium moniliforme*, *Candida albicans* was determined. Against both bacteria and fungi, the zone of growth inhibition and, thus, the antimicrobial activity was higher for biologically synthesized Ag NPs than for chemically synthesized Ag NPs [149]. Mohammad Musawi-Khattat et al. observed that besides higher antimicrobial activity of Ag NPs synthesized by green synthesis method compared to chemically synthesized Ag NPs, they also exhibited more desirable characteristics and biological activities such as narrow size range, spherical shape, high antioxidant and DNA cleavage activity [150]. Also in the case of gold nanoparticles, the green synthesis method results in higher antifungal activity and smaller size of the resulting nanoparticles compared to the chemical synthesis method [151]. Similarly, biosynthesized titanium and iron oxide nanoparticles were observed to have higher antimicrobial activity compared to chemically synthesized nanoparticles [152,153]. On the other hand, zinc nanoparticles synthesized by the chemical method had high thermal stability compared to ZnO NPs synthesized by the green synthesis method, while the antimicrobial activity was insignificantly higher for ZnO NPs prepared by the chemical method over by the green synthesis at 50 and 100 ppm, but no difference at 150 ppm against *Pseudomonas aeruginosa* and *Bacillus subtilis* with comparable activity against *Staphylococcus aureus* [154]. Recent studies on the antimicrobial activity of monometallic NPs are summarized in Table 1.

### 2.2. Metallic Nanoparticles in Combination with Antibiotics

Due to the antimicrobial activity possessed by metallic nanoparticles, they can overcome resistance mechanisms such as: (i) reduced permeability of bacterial cells, (ii) enzymatic modifications of antimicrobial substance, (iii) modification within target sites/enzymes, (iv) active removal of antimicrobials by overexpression of scavenger pumps to escape the antimicrobial effect of antimicrobials, or (v) overexpression of an enzyme inactivated by an antimicrobial [163,164,165]. Additionally, the coupling of metallic nanoparticles with antibiotics shows synergistic effects against bacteria in planktonic, as well as biofilm forms or also against multidrug resistant strains [166,167]. When combined with optimally selected antibiotics, nanoparticles exhibit synergy and in the future may contribute to the reduction of the global crisis of emerging microbial resistance [168]. The benefit of this combination is an increase in antibiotic or fungicidal activity due to a synergistic effect, resulting in a faster antimicrobial action and, thus, reducing the possibility of the emergence of resistant microorganisms, as well as an antimicrobial action against biofilm-forming pathogens and an increase in the penetration of antimicrobial agents into cells and tissues [169]. It is also worth pointing out that metallic nanoparticles do not have much potential for the induction of microorganisms resistance [58,142] and that antibiotic resistance is of little relevance to nanoparticles, because the action of nanoparticles takes place through direct contact with the cell walls of pathogens without the need to penetrate microbial cells, or use of the specific pathogen’s targets that might be modified by microbes in response to presence of nanoparticles [170].

Although broad-spectrum antibiotics and antifungal agents play a very important role in the control of bacterial and fungal infections, they also have a disadvantageous side to their use, namely, the selection and spread of resistance among many bacterial and fungal species and the deleterious effect they can have on the host microbiome [1,171,172,173]. Problems related to conventional antimicrobial therapy also include, but are not limited to, a narrow spectrum of antimicrobial activity, where the agent used is directed at a well-defined target of infection, or problems related to the safety and tolerability of the antimicrobial agent, which can cause harmful side effects such as toxicity or allergic reactions [174,175]. One of the major limitations of conventional antimicrobial therapy is also the inefficient delivery of drugs, where they may be non-specifically distributed in the body causing systemic side effects. In addition, there may be problems related to drug absorption and metabolism [176]. By increasing the potency of antibiotics by combining them with nanoparticles, it is possible to shorten the duration of treatment, reducing the concentration of administered drug to the patient, resulting in, among other things, in decreased systemic toxicity [177].

Colistin is considered as an antibiotic of last line of defense for the control of infections of some pathogens such as *Pseudomonas aeruginosa* resistant to all commonly used antimicrobial drugs. However, due to the dose-dependent side effects of colistin, the possibility of bacteria treating using colistin and seeking same therapeutic effect, but at a lower dose, is being sought. To achieve these goals, silver nanoparticles have been used in combination with colistin by Khaled et al. [178]. Additionally, the synergism of imipenem with Ag NPs was investigated. The synergistic effect of antibiotics with silver nanoparticles was determined against pandrug-resistant *Acinetobacter baumannii*. The results obtained indicate a synergistic effect leading to a reduction in the MIC values of colistin, imipenem and silver nanoparticles where a more than fourfold reduction was observed. Due to the synergistic effect of metallic nanoparticles with an antimicrobial agent, it is possible to target not only planktonic cells, but also cells growing within biofilm structure. Our research also confirms the synergism between metallic nanoparticles and antibiotics. As a result of the combination of the classical antibiotics such as vancomycin and colistin, synthetic ceragenins CSA-13 and CSA-131 and the human antimicrobial peptide cathelicidin LL-37 with core-shell magnetic nanoparticles against methicillin-resistant *Staphylococcus aureus* and *Pseudomonas aeruginosa*, an additive or synergistic effect was observed, as well as a strong suppression of biofilm formation. The interaction of magnetic nanoparticles with bacterial cell wall compounds results in increased insertion and/or uptake of membrane-active agents such as colistin or vancomycin, destruction of the membrane and leakage of intracellular contents, as well as induction of oxidative stress by the magnetic nanoparticles, causing damage to bacteria cell’ organelles [179]. Also, in the case of the combination of gold nanoparticles with tobramycin against tobramycin-resistant strains of *Pseudomonas aeruginosa*, a strong combinatorial effect of nanoparticles with an antibiotic was achieved, enabling the reduction of biofilm formation and, thus, increasing the effectiveness of antimicrobial therapies [121].

As a result of the combination of silver nanoparticles with amphotericin B and fluconazole, Ag NPs showed a synergistic effect with amphotericin B and fluconazole against biofilms formed by *Candida albicans*. As a consequence penetration of silver nanoparticles through the cell membrane due to their small size, the integrity of the membrane is disrupted, resulting in easy passage of drugs through the cell membrane leading to their action at the target site [180]. The metallic nanoparticles also showed high activity against fungal spores. Silver nanoparticles were synthesized by green synthesis using agro-waste material, strawberry leaf as reducing agents and completely large germination inhibition of *Botrytis cinerea* spores at 100 ppm as a result of the increased density of the solution, causing cohesion/sticking of the fungal hyphae [181]. In addition to the search for synergistic interactions between metallic nanoparticles and antimicrobial or fungicidal agents, other potential compounds are being explored to enhance the antimicrobial activity of the metallic nanoparticle-factor A complex. An example of this is the study by Al-Tawarah et al. of a synergistic interaction between silver nanoparticles and the essential oil of *Varthemia iphionoides*. The results showed a significant increase in antimicrobial activity of Ag NPs complex with essential oil against multi-drug resistant strains of *Enterobacter aerogenes*, *Pseudomonas aeruginosa*, *Staphylococcus epidermidis* and *Staphylococcus aureus*. The Ag NPs resulted in an increase in surface area, leading to greater surface contact with the bacteria and, thus, improved bactericidal activity, perforation and lysis of the bacterial cell wall, followed by generation of free radicals and DNA breakdown [182]. On the other hand, Abdelsattar et al. [183] evaluated the synergistic effect of silver nanoparticles with ZCSE2 phage against *Salmonella enteritidis*. Synergistically treating bacteria with a sublethal dose of Ag NPs enabled them to be readily lysed by phages even at low concentrations. As a result of the combination of Ag NPs and phages, a new prospect of nanoparticles with greatly improved antibacterial properties and therapeutic efficacy appeared.

Recent work on the synergistic effect of metallic nanoparticles in combination with antibiotics/fungicides and compounds other than antimicrobial agents is shown in Table 2.

### 2.3. Multimetallic Nanoparticles

Multimetallic NPs are nanoparticles composed of at least two different metals that form alloy or core-shells nanostructures. Multimetallic nanoparticles are of growing interest due to an increased spectrum of properties compared to monometallic NPs [191]. The bactericidal mechanism of action of multimetallic nanoparticles is usually related to the release of metal ions and the induction of oxidative stress, while non-oxidative mechanisms may also take place [191]. The joined action of various metals and metallic oxides in chemical transformation results in enhanced catalytic performance of multimetallic nanoparticles [192]. With regard to the synergistic effects between various metals, multimetallic NPs with bimetallic, ternary and quaternary combinations exhibit special features with improved chemical, optical and catalytic performance compared to mono- and bimetallic NPs [193].

By using a combination of metal compounds, it is possible to obtain synergistic antimicrobial properties of the newly synthesized compound compared to the properties of the individual components used alone. The antimicrobial activity of silver nanoparticles involves the anchoring and penetration of NPs in the bacterial cell wall while once inside the cell, they contribute to the formation of free radicals, generating intracellular oxidative stress and ultimately leading to cell death [194]. On the other hand, iron can interact with amino acids present in bacterial cell wall proteins, including the -SH groups of cysteine. The thiol side chain of cysteine has been shown to be the most susceptible to electron capture from oxidative species [195]. By synthesizing bimetallic silver and iron nanoparticles, Padilla-Cruz et al. [196] suggest that the mechanism of synergistic action of the two metals involved oxidation of the thiol side chains in cysteine leads to changes in protein structure, resulting in an increase in bacterial cell wall permeability and ultimately cell death (iron was responsible for this effect). As a result of the increased permeability of the cell wall, there is an increased influx of bimetallic nanoparticles into the cell. Eventually, with the release of silver ions into the cytoplasm, oxidative stress is induced, causing DNA changes and disruption of membrane morphology. In this way, synergistically acting silver and iron contribute to the destruction of cell structures, disruption of intracellular biological functions leading to cell death. In another study, Zhao et al. [197] noted that monometallic gold (Au), rhodium (Rh) and ruthenium (Ru) NPs did not cause disruption of bacteria cell structures of *Escherichia coli*, bacterial membranes treated with monometallic NPs had no visible damage. In contrast, the application of bimetallic gold- rhodium (Au-Rh) NPs and gold- ruthenium (Au-Ru) NPs caused significant changes in the cell membrane structure- cell membrane was dramatically ruffled and severely damaged, thus, can induce bacterial cell lysis, leading to leakage of cell substrates and bacterial death. The mechanism of action of the bimetallic nanoparticles also included a decrease in bacterial membrane potential and an increase in ATP and ROS levels. The above results suggest that monometallic nanoparticles (Au, Rh and Ru NPs) exhibit lower antimicrobial activity under the given experimental conditions (they do not cause noticeable changes in the cell membrane), compared to bi-metallic nanoparticles (Au-Rh and Au-Ru NPs), significant changes in the bacterial cell membrane were observed as a result of synergistic action of their constituent metals. Moreover, the application of bimetallic silver-platinum (Ag-Pt) NPs [198] not only killed the bacteria, but also limited their growth by reducing the density of bacteria, which shows that they are bacteriostatic agents; stopping bacteria from reproducing. The activity of multi-metallic nanoparticles covers a broad spectrum, not only against Gram-positive and Gram-negative bacteria, but also against multidrug resistant fungi such as *Candida auris*. Exposure of fungal cells to trimetallic silver-copper-cobalt (Ag-Cu-Co) NPs [199] influenced the level of apoptosis markers, manifested by phosphatidylserine translocation and collapse of mitochondrial membrane potential. The nanoparticles resulted in a direct inhibition of the cell cycle, arresting cells in the G2/M phase.

Some bacterial species have a remarkable ability to adapt to the administration of antibiotics by developing resistance mechanisms such as *Mycobacterium tuberculosis*, which is made possible by the rapid export of drugs from the cytosol. One of the targets of the silver and zinc oxide nanoparticles is to weaken the stability of the membrane, resulting in an increase in its permeability to antibiotics [200,201]. The use by Ellis et al. [202] of bimetallic nanoparticles in pulmonary delivery of antitubercular drugs to the endosomal system of *Mycobacterium tuberculosis*-infected macrophages in combination with rifampicin resulted in an increase in the potency of the antibiotic by as much as 76%, causing a decrease in the integrity of the *Mycobacterium tuberculosis* cell envelope due to the interaction of bimetallic nanoparticles with the mycobacterial envelope, which is reflected in an increase in its permeability. Due to this interaction, an increased penetration of rifampicin into the cytosol of the bacteria is possible, which results in an enhanced potency of the drug. The use of multi-metallic nanoparticles proved to be an effective drug delivery vehicle that can be used to transport TB drugs, among others, while increasing the potency of the drug [202].

Most bacterial infections are associated with biofilm formation, where the microbial cells that make up the biofilm structure have been shown to be 10–1000 times more resistant to antibiotics than planktonic cells [203]. As a result, it is necessary to develop new bactericides that can effectively combat biofilm-associated infections. One example of such agents represents the silver-platinum nanohybrids synthesized by Ranpariya et al. [204], which significantly inhibited bacterial biofilm formation and exhibited strong antimicrobial synergy when combined with antibiotics such as streptomycin, rifampicin, chloramphenicol, novobiocin, and ampicillin against strains of *Escherichia coli*, *Pseudomonas aeruginosa*, and *Staphylococcus aureus*. For example, they found that rifampicin activity in the presence of Ag-ZnO NPs increased as much as 15-fold against *Staphylococcus aureus*, while Ag-ZnO NPs inhibited biofilm against *Escherichia coli* and *Pseudomonas aeruginosa* by about 76%. Bimetallic nanoparticles Ag-Au NPs synthesized by the core-shell method [205] showed synergistic antimicrobial activity of bimetallic nanoparticles conjugated with doxycycline against *Pseudomonas aeruginosa* and *Escherichia coli*, where the combinatorial effect led to higher drug binding affinity and enhanced antimicrobial efficacy. Synergy of antibiotic with bimetallic nanoparticles may be the current approach with the most promise for the significant improvement of patients treatment with complicated skin infections.

Recent studies on the antimicrobial activity of multi-metallic NPs are summarized in Table 3. Quadrometallic nanoparticles, such as silver-copper-platinum-palladium (Ag-Cu-Pt-Pd) [206] or silver-platinum-gold-palladium (Ag-Pt-Au-Pd) [207], are also being synthesized, but so far no studies have been conducted on the antimicrobial properties of quadrometallic nanoparticles.

As a result of the generation of reactive oxygen species, the antioxidant defense system is disrupted, which leads to mechanical damage to the cell membrane. A large number of studies on multimetallic nanoparticles describe their mechanism of action as the adhesion of multimetallic NPs to microbial cells and destruction of the cell wall by interaction between the positively charged surface of multimetallic NPs and the negatively charged surfaces of pathogen cells, leading to the generation of ROS, the penetration of multimetallic NPs into the cell, causing damage to proteins and DNA, as well as oxidative stress [191]. Considering the wide spectrum of action of multimetallic nanoparticles, and the diversity of their mechanisms of action against pathogens, including multidrug-resistant strains, they may prove to be an effective tool to combat infections.

### 2.4. Metallic Nanoparticles as Carriers for Molecules with Antimicrobial Activity

Metallic nanoparticles have great potential in medicine as carriers of small molecules such as drugs, genes, proteins, and enzymes [215,216,217]. The efficacy of some antibiotics can be enhanced by increasing the cell permeability or weakening the cell envelope, therefore, when metal nanoparticles are combined with antibiotics, they can show better efficacy in certain therapies by reducing the side effects associated with individual drug [218]. Functionalization of antimicrobial agents with nanoparticles is one of the strategies used to enhance the efficacy of drugs against pathogens.

Functionalization of metallic nanoparticles with antimicrobial agents enables strong antimicrobial activity as a result of enhanced ability to penetrate biological membranes. Penetration of hydrophobic antimicrobials is limited by the highly polar environment within bacterial membranes, which impairs their activity [73]. Metallic nanoparticles interact with the bacterial cell membrane through electrostatic, hydrophobic, receptor-ligand interactions or van Der Waals forces, leading to a change in the cell membrane potential and bacterial integrity [219,220]. Due to the high surface-to-volume ratio and the possibility of loading the metallic NPs onto the surface with a high concentration of antimicrobial agent, increased permeability towards the biological membrane or higher uptake by the bacterial cell, the effective delivered concentration of antimicrobial agent is increased [221,222]. As a result of the increased porosity of the pathogen’s structure, antibiotic molecules conjugated with metallic NPs gained easy access to the bacterial cell. The hypothesis proposed by Shaikh et al. is that conjugation on the surface of nanoparticles can result in increased concentrations of the administered antibiotic, which is able to saturate antibiotic-degrading enzymes and inhibit the growth of resistant bacterial strains containing degrading enzymes [222]. On the other hand, Sreedharan et al. proposed a hypothesis regarding the increased permeation of ciprofloxacin (AuF NPs@ciprofloxacin) bound on the surface of gold nanorods by: (i) the binding of AuF NPs@ciprofloxacin to the cell wall or membrane of the microorganism resulting in the release of carried drug within the cell wall or membrane or (ii) the nanoparticle-antibiotic complex binds to the bacterial cell wall that may serve as a reserve for the continuous release of the antibiotic, which could then penetrate into the microorganisms [223]. In addition, efflux pumps, whose activity is increased in antibiotic-resistant bacterial cells, play an important role in multidrug resistance, whereby antimicrobials are actively transported outside the bacterial cells [224]. As a result of the functionalization of metallic nanoparticles with agents with antimicrobial activity, it is possible to block the efflux pump, increasing the accumulation of antibiotics inside bacterial cells [225]. Brown et al. showed that gold nanoparticles functionalized with ampicillin were observed to block the efflux pump and the multivalent presentation of ampicillin were the reason for the more effective action of the functionalized gold nanoparticles versus the antibiotic compared with the antibiotic alone [226].

Another advantage of the functionalization of metallic nanoparticles is the improved stability of the metallic NPs-antimicrobial agent. Higher stability and antimicrobial activity under conditions such as room temperature, UV exposure or heat stress (increasing temperature up to 90 °C) was reported for the conjugate of Au NPs with ampicillin, streptomycin and kanamycin compared to the free forms of the antibiotics (except for Au NPs@ampicillin, where at room temperature the conjugated ampicillin was precipitated out of solution) [227]. Metallic nanoparticles are also a good carrier, providing high antimicrobial peptide (AMP) activity in the presence of proteases and enzymes [228]. The stability of metallic nanoparticles in different buffer solutions and biological fluids such as water, Dulbecco phosphate buffered saline (DPBS, in different pH range) in various concentrations of NaCl and in the presence of fetal bovine serum confirms that they are a promising approach in drug delivery [229].

An important aspect of antimicrobial drug delivery based on metallic nanoparticles is the improvement in the pharmacokinetic properties of the drug in the form of increased solubility of poorly soluble drugs, prolonged drug half-life and systemic circulation time, as well as prolonged and stimulus-controlled drug release, resulting in lower dosage and drug frequency, reducing the toxic effect of the drug [163]. By functionalizing the surface of metallic nanoparticles with antimicrobial agents, it is possible to overcome their poor solubility and aggregation in solution, thereby achieving an increase in antimicrobial efficacy and a decrease in cytotoxicity [230,231]. Metallic nanoparticles as carriers for antimicrobial agents can protect drugs from premature degradation and sustain drug release in order to result in prolonged half-life and bioavailability [232]. The conjugation of gentamicin with gold nanoparticles confirms that metallic nanoparticles are a very good carrier for continuous release of the antibiotic over a few days, making it possible to reduce the number of administrations [233]. By coupling antimicrobial agents to metallic nanoparticles, it is also possible to improve antimicrobial properties and overcome resistance mechanisms among microorganisms. Our research confirms that metallic nanoparticles are an effective carrier for antimicrobial agents. When conjugated to gold nanoparticles with ceragenins, they show higher antimicrobial activity against both multidrug-resistant strains regardless of resistance mechanism [117], strains causing otitis media [119] and fungal strains [142]. In addition, the magnetic nanoparticles prove to be a very good carrier for the PBP10 peptide, which shows good antimicrobial activity against both planktonic and biofilm forming of bacteria and fungi [143]. Also, the conjugation of 1,4-dihydropyridine on the surface of magnetic nanoparticles significantly increases antimicrobial activity compared to nanoparticles alone which is due, among other things, to the high affinity of the nanosystems for microbial cell wall components, while antimicrobial activity is still high in the presence of human body fluids such as serum, saliva, cerebrospinal fluid or abdominal fluid [144]. A microorganism that is originally resistant to a given antimicrobial agent becomes susceptible to the nanosystems conjugated with metallic nanoparticles. Carbapenem-resistant *Acinetobacter baumannii* was found to be sensitive to conjugated Ag NPs with imipenem [234], where were observed (i) reversal of drug resistance by protecting the β-ring of carbapenem from hydrolysis by metallo-β-lactamases (MBLs) through zinc ion chelation of MBLs, resulting in the deactivation of MBLs, and (ii) enhanced antibacterial efficacy with increased production of reactive oxygen species and membrane damage, (iii) effects on cell wall formation and metabolic pathways, as well as the downregulation of ompA gene expression, which can mediate fibronectin-mediated attachment to host cells and induce the biofilm formation. Despite the presence of beta-lactamase and carbapenemase resistance genes in *Acinetobacter baumannii*, the combination of AgNPs with imipenem is effective antimicrobial agent against carbapenem-resistant strains, showing potent antimicrobial activity [235]. Ampicillin-resistant *Escherichia coli* also proved to be sensitive after exposure to ampicillin-conjugated gold nanoparticles, where an accumulation of Au NPs@ampicillin on the bacterial cell surface was observed, resulting in the formation of pores in the bacterial membrane, allowing the nanoparticles to penetrate the interior of pathogen cells [236]. The results obtained by Memarian et al. confirm that the fluconazole-resistant strain became sensitive to Au NPs@fluconazole. The MIC value for fluconazole alone was 64 µg/mL, while that for the tested nanosystem was 2 µg/mL [237].

Another advantage of functionalizing metallic nanoparticles with antimicrobial agents is the reduction of toxic effects where using nanoparticles as carriers for drug delivery not only improves efficacy, but it also enables a reduction in adverse effects compared to conventional therapy. Functionalization of gold nanoparticles with ciprofloxacin resulted in lower hemolytic activity of Au NPs@ciprofloxacin than the antibiotic in free form thereby reducing the toxicity of the antibiotic [238]. Similarly, conjugation of amphotericin B (AMB), which exhibits nephrotoxicity due to its poor water solubility and aggregation on the surface of gold nanoparticles allowed the negative effects to be reduced, resulting in a water-soluble covalent gold nanoparticle conjugate with AmB with increased antimicrobial efficacy and reduced cytotoxicity [230]. Our studies also confirmed that conjugated ceragenins, peptide LL-37 chlorhexidine and polyene antibiotics (amphotericin B and nystatin) on the surface of magnetic nanoparticles showed not only lower toxicity, but also increased antimicrobial activity compared with antimicrobial agents in a free form, which is a very promising approach to reduce the side effects of conventional therapies and increase the success of therapies [145,146,147,239].

By functionalizing metallic nanoparticles with specific antibodies, it is possible to obtain a system that might serve for rapid identification of the pathogens and target treatment to combat infection. Gold and silver nanoparticles conjugated with antibodies specific for *Staphylococcus aureus* peptidoglycan are proving to be a promising treatment method, which can be used alone or in addition to existing conventional antibiotic therapy to achieve complete eradication of the pathogen by means of which extended and selective bacterial death can be achieved [240,241,242]. Also, the conjugation of antibodies to protein A on the surface of gold nanoparticles both in vitro and in vivo in a mouse model resulted in a significant reduction in the viability of methicillin-resistant *Staphylococcus aureus* (MRSA) cells and the ability of the antibody-nanoparticle conjugate to selectively kill pathogens in an animal model [243]. Goat anti-*Escherichia coli* O157:H7 antibodies were also successfully conjugated on the surface of silver nanoparticles, which effectively bind to the target pathogen [244]. An important aspect of the successful treatment of infections is the correct identification of the pathogen. Hashemi et al. confirmed that with rabbit antibodies to *Candida* and *Gardnerella* species, it is possible to correctly identify vaginal infections with very high sensitivity and specificity [245].

Recent studies on the antimicrobial activity of metallic NPs functionalized with antibiotics/fungicides or compounds other than antimicrobial agents are summarized in Table 4.

## 3. Biocompatibility of Metallic Nanoparticles

An important aspect of the biomedical application of various types of nanoparticles is biocompatibility, i.e., the property of a substance determining its correct functioning in a living organism, which should show a lack of toxicity, not affect the body’s immune system and do not induce hemolysis. In order to ensure the effective and safe use of nanomaterials, the interactions between the nanoparticle and the cells of the host must be considered, with particular attention being paid to the environment in which the test compounds act. A great number of information/trends regarding the toxicity of nanoparticles are obtained in cancer research and in studies that are not focused on antimicrobial activity. On the basis of such studies, certain effects can be expected, even their possible therapeutic potential [254,255]. The toxicity of nanoparticles is highly affected by their physical and chemical properties, such as shape, size, surface area and charge or catalytic activity [256].

Therefore, different methods are used to assess the biocompatibility of nanoparticles, ranging from quantitative assays using conversion of compounds such as 3-(4, 5-dimethylthiazol-2-yl)-2, 5-diphenyltetrazolium bromide (MTT) [257], 2, 3-bis-(2-methoxy-4-nitro-5-sulfophenyl)-2H-tetrazolium-5-carboxanilide (XTT) [258] or lactate dehydrogenase (LDH) [257], through qualitative studies (live and/or dead cell staining assays using dyes such as calcein-AM with propidium iodide [259] and dual acridine orange/ethidium bromide staining [260]) and finally blood hemolysis assays [261] and animal model [262] to assess biocompatibility.

A very important factor associated with the toxic properties of metallic nanoparticles is their size. Due to their small size, NPs have a much larger surface area per unit mass compared to their bulk counterparts, which translates into higher reactivity, which is associated with a higher risk of cytotoxic effects. As their size decreases, the number of metal atoms per surface area increases exponentially, resulting in higher reactivity of nanoparticles in biological systems [263]. Due to their small size, many nanoparticles are able to bypass or cross the blood-brain barrier where they can reach and accumulate in the brain parenchyma, including the striatum and hippocampus [264]. Depending on the size of the nanoparticles, a differentiated subcellular distribution is observed in the accumulated organs. In a study provided by Lopez-Chaves et al. gold nanoparticles with three sizes of 10 nm, 30 nm, 60 nm were observed to accumulate. Au NPs of 10 nm in size gathered within the cell nucleus, while particles larger than 10 nm in the cytoplasm [265]. Xia et al. determined the effect of cytotoxicity of gold nanoparticles depending on their size (5, 20 and 50 nm) against HepG2 cancer cells and healthy L02 cells, where Au NPs of 5 nm size showed higher cytotoxicity than those of 20 and 50 nm size. In contrast, in mouse in vivo studies, 50 nm Au NPs showed the longest circulation in the blood and the highest distribution in the liver and spleen, while 5 nm Au NPs caused an increase in neutrophil counts and little hepatotoxicity in a mice [266]. In addition, the size of metallic NPs may determine the aggregation process. Results presented by Wozniak et al. showed that 4–28 nm Au nanospheres aggregate at high concentrations and long incubation times increasing cytotoxicity in contrast to larger 130 nm star-shaped Au nanoparticles, which are rather monodisperse and non-toxic [267].

With the help of changing the shape of nanoparticles, it is possible to modulate their cytotoxicity. Nanoparticles can have different shapes and geometries including spheres, ellipsoids, cylinders, stars, octahedral sheets, cubes, spikes, rods, triangles, prisms, which significantly affects their toxicity. The star-shaped AuNPs had the highest anticancer potential but also exhibited the highest cytotoxicity, while the spheres of AuNPs, which were the least cytotoxic, showed weak anticancer activity [268]. Lee et al. synthesized chitosan-coated gold nanoparticles in the shape of nanospheres, nanostars and nanowires and determined the effect of their shape on cytotoxicity against human hepatocyte cancer cells (HepG2). Cytotoxicity was highest for Au NPs in the shape of nanorods, followed by nanostars and lowest for nanospheres [269]. Whereas Wozniak et al. performed the synthesis of Au NPs with different shapes: spherical (~10 nm), nanorods (~41 nm), nanoprisms (~160 nm), nanostars (~240 nm) and nanoflowers (~370 nm) against cancer cells—HeLa and normal cells—HEK293T. The obtained results indicated that spherical and rod-shaped Au NPs were found to be more toxic than star-, flower- and prism-shaped Au NPs, which the authors suggest may be due to the aggregation process and their small size. The above results indicate that the selection of the appropriate shape for the synthesis of metallic nanoparticles affects their cytotoxic activity [267]. Our results provided information that ceragenin-functionalized gold nanoparticles (CSA-13, CSA-44 and CSA-131) with the peanut-shape induce the greatest hemolysis compared with rod- and star- shaped [117,142], while non-functionalized gold nanoparticles in rod-shaped induced slightly greater hemolysis compared to peanut-shaped Au NPs.

The toxicity of nanoparticles may also depend on their chemical composition. The degradation of nanoparticles that can occur depends on environmental conditions, such as a change in pH, ionic strength, or ionic valence, resulting in the leakage of metal ions from the core of metallic nanoparticles [270]. The resultant release of metal ions, such as silver, cobalt, chromium, or nickel, is toxic to cells and causes cell damage, whereas the release of metal ions from the nanoparticles alters the bioactivity and thus the toxic effect of the nanoparticle-metal ion complex [271]. Free ions can cause, among other things, oxidative stress with the release of cytokines [272]. In turn, other metal ions such as iron or zinc, which are the main micronutrients necessary for the proper functioning of the body, however, as a result of exceeding a certain concentration can adversely affect the functioning of cells by negatively affecting cellular pathways and thus cause high toxicity. The toxic effect of released metal ions can be reduced by using, among other things, appropriate surface modifications, thanks to which their properties can be improved and such system can be stabilized by preventing the release of ions from the interior, preventing oxidation of nanoparticle surface and inhibiting aggregation and subsequent agglomeration of nanoparticles [273]. Results obtained by Soenen et al. indicated that coating silver nanoparticles with a thin layer of SiO_2_ minimized their toxicity as a result of blocking ion release and bacterial and/or cell contact. In addition, the composition of the core can be changed by the addition of other metals, thereby achieving increased chemical stability against degradation of the metallic nanoparticles and consequently against unwanted leakage of metal ions into the body [274].

The surface charge of nanoparticles plays an important role in their toxicity, as it determines to a large extent the interaction of nanoparticles with biological systems [275]. The relationship between high toxicity and positive charge on the surface of NPs is explained by their ability to penetrate into cells resulting from electrostatic interactions between negatively charged cell membrane glycoproteins and positively charged NPs, where in the case of neutrally or negatively charged NPs such interactions are not observed [256]. The surface charge of metallic nanoparticles can be modified by non-covalent modification of the nanoparticle surface by coating or wrapping with biological molecules to make more biocompatible NPs using polymers, peptides, proteins, or surfactants. The second way to modify the surface charge are covalent modifications involving the formation of chemical bonds between functional groups present on the nanoparticle surface and other biological molecules attached to that surface such as polyethylene glycol, peptides, or carbohydrates [276]. Chen et al. performed the synthesis of copper oxide nanoparticles modified with the polymers polyethylene glycol (PEG), polyvinylpyrrolidone (PVP), polydopamine (PDA) and polyvinyl alcohol (PVA) to determine the effect of surface modification of nanoparticles on their antimicrobial activity against *Escherichia coli*. The positive surface charge of CuO-PVP NPs enhanced antibacterial activity through electrostatic interactions with negatively charged surfaces of *Escherichia coli*. The authors concluded that the positive surface charge of CuO-PVP NPs resulted in enhanced antibacterial activity through electrostatic interactions with negatively charged *Escherichia coli* surfaces. It was also observed that the weakly negatively charged CuO-PDA NPs achieved better antibacterial activity, which the authors explain by the presence of lipophilic catecholamine structures on the nanoparticle surfaces, which enabled interaction with the lipid bilayer in the outer membrane of *Escherichia coli* [277]. In addition, Abbasadegan et al. synthesized silver nanoparticles, obtaining three different electrical surface charges: positive, neutral and negative and determined the antibacterial activity of the nanoparticles. The authors concluded that the surface charge of Ag NPs was a significant factor affecting the bactericidal activity, where positively charged nanoparticles showed the highest bactericidal activity against both Gram-positive and Gram-negative bacteria, negatively charged nanoparticles showed the least antibacterial activity and neutral nanoparticles had intermediate activity [56].

The biocompatibility of metallic nanoparticles is also influenced by the method of synthesis and its conditions. Using the MTT assay, Amooaghaie et al. determined the toxicity of silver nanoparticles synthesized by two methods: green synthesis using *Nigella sativa* extract and chemical synthesis against bone-building stem cells of mice. The toxicity of the green synthesized Ag NPs was significantly lower than that of the chemical synthesized Ag NPs after 24, 48 and 72 h. After 72 h exposure of cells to the test compounds at a concentration of 0.2 mg/L, more than an 11-fold decrease in the number of viable cells was observed for chemically synthesized Ag NPs compared to Ag NPs synthesized by the green synthesis method [278]. Ghetas et al. determined the toxicity of chemically and biologically synthesized silver nanoparticles by means of a hemolysis assay using on chicken and goat red blood cells. The results indicate that chemically synthesized Ag NPs are in most cases more hemolytic than biologically synthesized Ag NPs [149]. Similarly, in the case of FeO NPs, the nanoparticles obtained by green synthesis are more biocompatible than the counterpart synthesized by the chemical method [279]. Slightly different results were obtained by Kummara et al. where, following exposure of non-small cell lung cancer cells (NCI-H460) and normal human skin fibroblast cells (HDFa) to silver nanoparticles by green synthesis and chemical methods, these biosynthesized Ag NPs were found to be more toxic than chemically synthesized Ag NPs. Both lower cell viability and greater inhibition of the cell proliferation were observed when exposed to green synthesized Ag NPs [280].

## 4. Metallic Nanoparticles—Development of Microbial Resistance and Their Impact on the Host Microbiome

Due to the rapid spread of resistance among microorganisms [281], a very important aspect of the potential use of metallic nanoparticles is whether pathogens can become resistant to them, and how quickly. Due to the different mechanisms of action of nanoparticles, pathogens do not easily acquire resistance with regard to the need to develop multiple mutations [52,99]. There are reports that metallic nanoparticles do not induce the development of resistance. Xie et al. synthesized quaternary gold nanoclusters coated with quaternary ammonium and did not observe an increase in the MIC for *Staphylococcus aureus* after 30 days of exposure [58]. Zheng et al. also observed no change in MIC values after 30 days of passaging *Staphylococcus aureus* with cercaptopyrimidine-conjugated gold nanoclusters. The results of our induction of resistance in *Candida* strains exposed to ceragenin-functionalized gold nanoparticles over 25 passages also confirmed the low potential for resistance development among fungi [142]. On the other hand, another published study investigated whether microorganisms may develop defense strategies to cope with the antimicrobial effects of metallic nanoparticles. Adaptive defense mechanisms include reduced uptake/adsorption of metallic NPs, where an important role is played by porins involved in the transport of ions through outer membranes to the periplasmic space, from which they undergo specific transport across the cytoplasmic membrane to the cell interior. As a result of the down-regulation of porins, it is possible to reduce the destructive effect of metal ions and, thus, pathogens become less susceptible to metallic nanoparticles [282,283]. Another defense mechanism is the increased pumping of metal ions to the interior of the cell. Thus, the exposure of pathogens to metallic nanoparticles might results in the upregulation of genes encoding efflux pumps [282,284]. It should be noted that due to the upregulation of genes encoding a wide variety of efflux systems, it is also possible to remove antibiotics from inside of the cell as well, and therefore develop resistance not only to metallic nanoparticles but also to other antimicrobial agents. The enhanced detoxification of reactive oxygen species is also one mechanism. As a consequence of the exposure of pathogens to metallic nanoparticles, an increase in the expression of genes encoding ROS scavenging systems has been noted [285,286]. Furthermore, as a result of biofilm formation, a physical barrier is formed which impedes the penetration of metallic nanoparticles so that pathogen cells are exposed to lower doses of nanoparticles and are able to become resistant to the acting agent [287]. In respect of the influence of sublethal doses on the bacterial biofilm, Ouyang et al. [288] and Yang et al. [289] concluded not only the induction of quorum sensing gene expression and LPS (lipopolysaccharide) biosynthesis, but also the release of signaling molecules by *Pseudomonas putida* and *Pseudomonas aeruginosa* PAO1, respectively.

Another important aspect of metallic nanoparticles including, Ag NPs, TiO_2_ NPs and ZnO NPs, is that they potentially interfere with the intestinal microbiome that can compromise the host health [290,291]. At the beginning, it should be mentioned that NPs may affect a complex of gastrointestinal (GI) environment. The non-absorbed fractions of NPs accumulate in the intestine and can indirectly affect the intestinal microbiota occurring within the gut lumen, as well as the mucus layer lining the epithelial surface. Subsequently, a portion of the NPs may translocate via the epithelial barrier and can be potentially captured by the intestinal immune cells (e.g., macrophages and dendritic cells) until reaching systemic circulation [290]. It should be explicitly underlined that intestinal microbiota plays a crucial role in numerous physiological functions as an indispensable element for host health. Apart from their contribution in the digestion of dietary fiber or the production of essential metabolites for the host, gut microbiota also participates in the maintenance of structural integrity of the mucosal barrier, the control of the immune response, and the protection against pathogens [290,292]. The intestinal microbiota consist of trillions of microorganisms comprising bacteria, viruses, fungi, archaea and protozoa [290,291,293]. It is estimated that in adults, the gastrointestinal tract (GIT) tract harbors 100 trillion bacteria, involving a minimum of several hundred species and more than 7000 strains [290]. Notably, 80% of fecal microbiota in a healthy adult representing the three dominant phyla i.e., Bacteroidetes, Firmicutes, and Actinobacteria [290,291]. Whereas other species are classified to the phyla Proteobacteria, Verrucomicrobia, Fusobacteria and Cyanobacteria [290]. A summary of the findings of research evaluating the impact of NPs on the intestinal flora is presented in Table 5.

Considering all data, it should be highlighted that Ag NPs, TiO_2_ NPs and ZnO NPs may affect the intestinal microbiota including the alteration of the F/B ratio (for example, an enhanced F/B ratio is related with obesity), a depletion of *Lactobacillus* strains and an elevated in the abundance of Proteobacteria. The above consequences can lead to obesity or even CRC (colorectal cancer) where gut dysbiosis play a significant role [290]. What is more, dysbiosis as a result of the action of the mentioned NPS may be associated with the development of inflammatory bowel disease (IBD), irritable bowel syndrome (IBS), and metabolic syndrome [290,291,292]. With regard to the above insights, additional investigations are needed for a better understanding of the changes among intestinal microbiota in the presence of Ag NPs, TiO_2_ NPs and ZnO NPs.

## 5. Conclusions and Future Perspectives

In an era of increasing antimicrobial resistance, metallic nanoparticles appear to hold promise to improve current therapies and to develop new therapeutic agents of nanoscale nature. A possible way to increase the antimicrobial activity of conventional antibiotics is to use them in combination with metallic nanoparticles. This can be done either by synthesizing metallic nanoparticles with a modified surface, functionalization with suitable antimicrobial agents, or by designing a core of synergistically acting metals. It is also possible to select antimicrobial agents and use them in combination therapy (metallic nanoparticles + antimicrobial agents). The use of synergism in the nanotechnology context seems to be a very promising approach in the fight against various infections, whether of bacterial or fungal etiology, and against planktonic and biofilm-forming bacteria. Thus, the synergistic combination of metallic NPs with suitable compounds may be a potential source of alternative antimicrobial agents and may play a significant role in the near future. Combining metallic nanoparticles with an antimicrobial agent not only demonstrated synergistic effects, but also improved drug delivery and increased antimicrobial agent efficacy, while reducing the side effects associated with the broad use of these agents. In addition, the synergistic action of metallic nanoparticles with an antimicrobial agent enabled resistance mechanisms to conventional antibiotics/fungicides, thereby providing a more effective use of antibiotics/fungicides available in the clinical practice.

In addition, the activity as well as the toxicity of metallic nanoparticles is influenced by their physical and chemical properties, where the size of nanoparticles has been found to have a significant effect due to the increase in specific surface area at nanoscale sizes, resulting in a higher number of cellular interactions and, consequently, toxicity. Other key factors influencing nanoparticle cytotoxicity include shape, surface charge, method of synthesis, nanoparticle surface modification and metal ion release. Therefore, the control of the physicochemical properties of NPs is crucial in order to obtain safer and more stable NPs. An important aspect of the use of nanotechnologies is the biocompatibility of the design materials. Although metallic nanoparticles in combination with antimicrobial agents have many advantages, they can also have adverse effects, as they have the potential to cross the natural barriers of living cells and tissues, causing toxic and inflammatory responses. The biocompatibility of metallic nanoparticles can be modulated by changing the size of the nanoparticles. It has been found that due to their small size, NPs have a much larger surface area per unit mass compared to their bulk counterparts, resulting in a greater reactivity with a greater risk of cytotoxic effects. By changing factors such as shape, chemical composition, surface charge, and method of synthesis, the biocompatibility of metallic NPs can be influenced. Therefore, it is believed that by controlling the physicochemical properties of nanoparticles, safer and more reliable nanoparticles with high antimicrobial activity can be obtained. Further research is still required to determine how nanoparticles affect the complex human body. In order to reduce the toxic effects of metallic nanoparticles, as well as in combination with other compounds, new methods are needed to reduce the negative effects of NPs while maintaining their activity. Much of the work to date confirms the low potential of metallic nanoparticles due to their divergent mechanisms of action against pathogens. However, there are also reports of the possibility of the development of resistance mechanisms as a result of exposure to nanoparticles, so a highly effective antimicrobial agent such as metallic NPs should be handled rationally to remain effective against various types of infections. Currently, the impact of NPs on intestinal microbiota has caught the increasing attention of researchers. It should be emphasized that there is a compelling amount of evidence indicating the correlation of Ag NPs, TiO_2_ NPs and ZnO NPs in the development of obesity, IBD, IBS metabolic syndrome or even CRC. Nevertheless, the above mentioned findings raise the need for extended investigation in order to fully explain them, especially in humans.

For the safe use of metallic nanoparticles there is a need for future research, which should include the mechanisms of NPs’ translocation, accumulation, long-term and long-lasting effects on the body, their interaction with cells, signaling pathways and receptors, and effects on fundamental processes such as phagocytosis. Understanding the relationship of potentially new nanotechnology-based antimicrobials to biological systems is a key to overcoming the limitations and toxicity barriers of metallic nanoparticles and to their future use in the treatment of bacterial or fungal infections. Although the results presented in this review are promising in the context of fighting infections by combining metallic nanoparticles with antimicrobial agents, at present, no existing product has been approved by the Food and Drug Administration (FDA) for clinical use. Therefore, future research should focus on the elucidation of the interaction of nanotechnology-based antimicrobials with cells, tissues or organs, their metabolism and accumulation in the body, and their effect on the natural flora in order to produce safe and effective means of combating microbial resistance in the next generation of drugs.

## Figures and Tables

**Figure 1 ijms-24-02104-f001:**
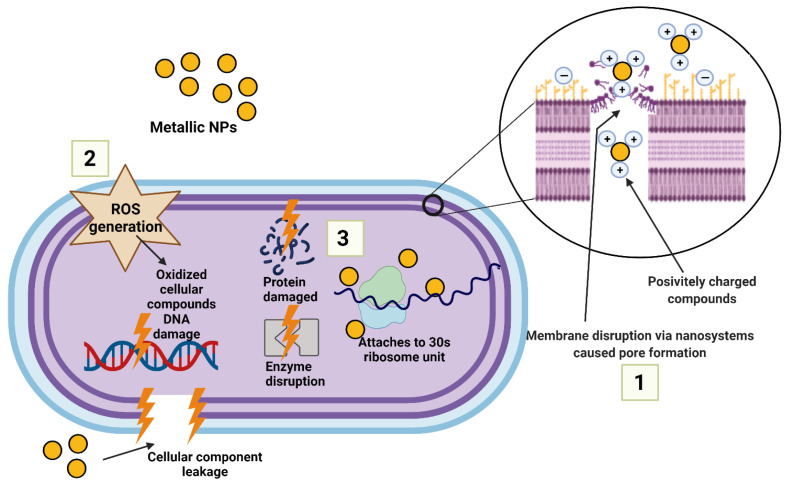
The main mechanisms of antimicrobial activity of metal nanoparticles include: (1) disruption of the pathogen cell wall resulting in increased permeability, (2) generation of ROS disrupting redox homeostasis and damaging cellular structures, (3) binding to intracellular structures causing their dysfunction.

**Figure 2 ijms-24-02104-f002:**
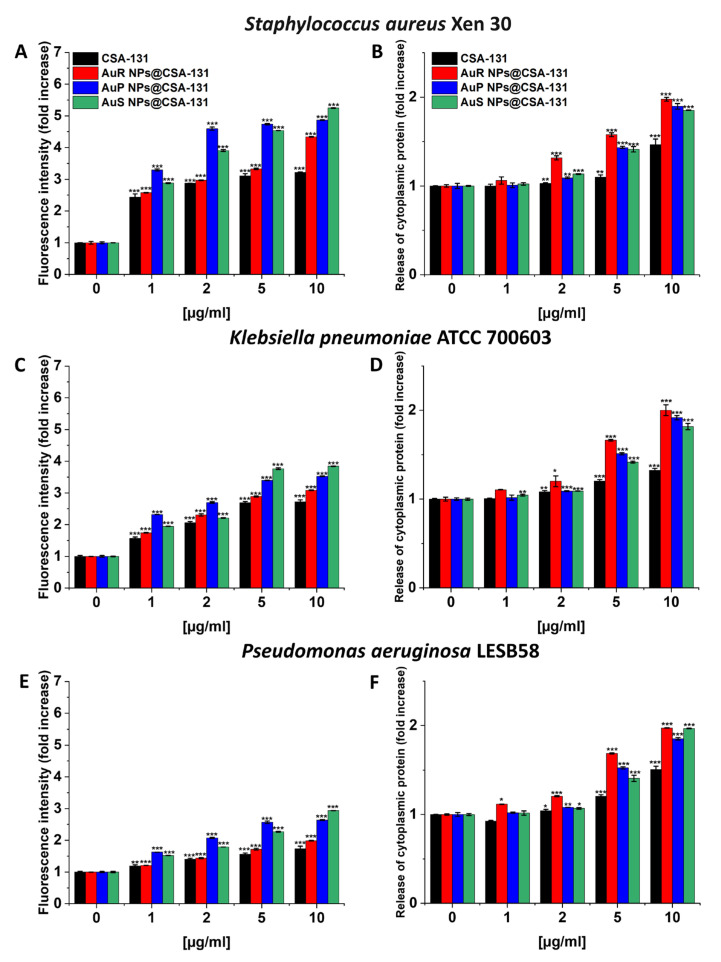
Bacterial membrane depolarization of multidrug-resistant strains: *Staphylococcus aureus* Xen 30 (**A**), *Klebsiella pneumoniae* ATCC 700603 (**C**), *Pseudomonas aeruginosa* LESB58 (**E**), was assessed using the 3,3′-dipropylthiadicarbocyanine iodide (diSC_(3)_) assay, where bacterial cells were treated with gold nanosystems functionalized with ceragenin CSA-131 with rod-shaped (AuR NP@CSA-131), peanut-shaped (AuP NP@CSA-131), and star-shaped (AuS NP@CSA-131) metal cores and free ceragenin CSA-131. The release of cytoplasmic proteins from the bacteria *Staphylococcus aureus* Xen 30 (**B**), *Klebsiella pneumoniae* ATCC 700603 (**D**), *Pseudomonas aeruginosa* LESB58 (**F**), treated with AuR NP@CSA-131, AuP NP@CSA-131, AuS NP@CSA-131 and CSA-131 was assessed using the Bradford protein assay. Concentrations of the tested compounds ranged from 1–10 µg/mL. Results show the mean ± SD, n = 3; * indicates statistical significance at p ≤ 0.05, ** ≤0.01, and *** ≤0.001. Adapted from Pharmaceutics [117].

**Figure 3 ijms-24-02104-f003:**
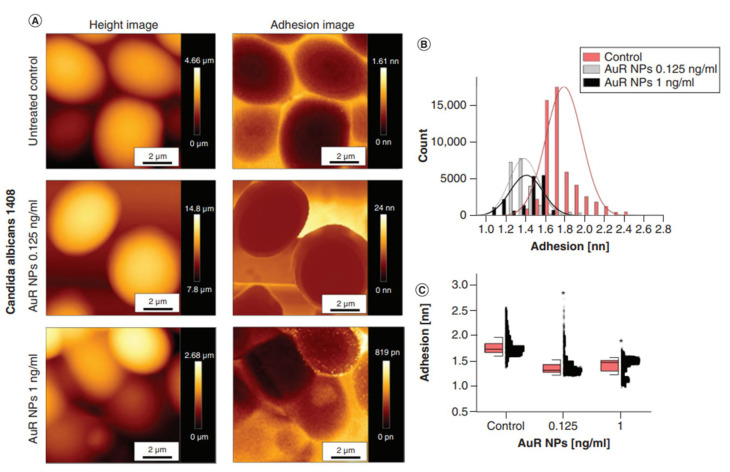
Height and adhesion images of *Candida albicans* strain 1408 using atomic force microscopy (**A**). Figures (**B**,**C**) show the adhesion distribution of *Candida albicans* strain treated with rod-shaped gold nanoparticles at doses of 0.125 ng/mL and 1 ng/mL compared to control cells. * indicates statistical significance (*p* < 0.05) compared to untreated control. Reprinted with permission from [120]. 2022, Medical University of Bialystok.

**Figure 4 ijms-24-02104-f004:**
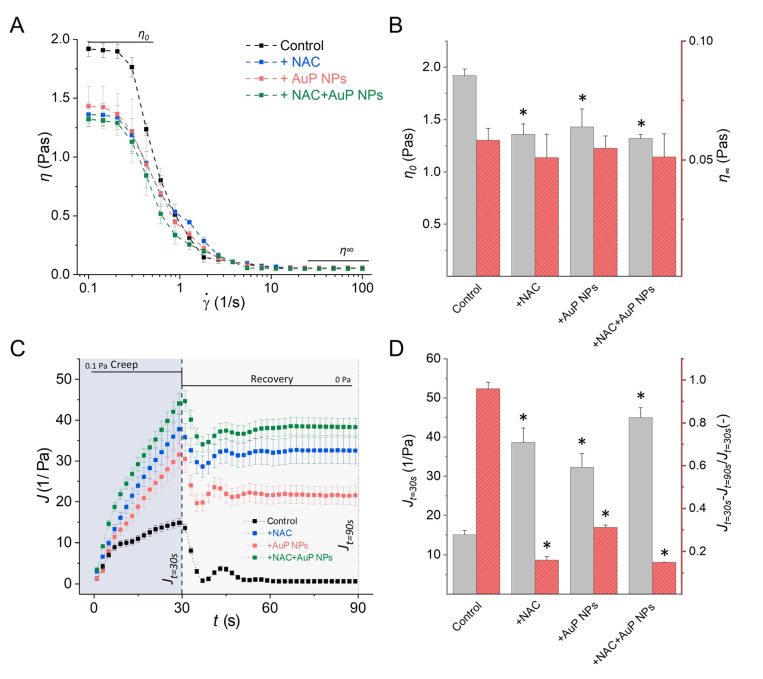
Rheological properties of *Pseudomonas aeruginosa* biofilm under the influence of N-acetyl-cysteine (NAC) and peanut-shaped gold nanoparticles (AuP NPs): (**A**) dynamic viscosity as a function of shear rate for control samples, and samples treated with tested compounds; (**B**) zero-shear viscosity η0 and infinity-shear viscosity η∞ determined from the viscosity curves; (**C**) compliance as a function of time in creep-recovery tests; (**D**) mean maximal creep compliance values (at 30 s Jt = 30 s) and the ratio of difference between Jt = 30 s and unrecovered creep compliance (at 90 s Jt = 90 s) to maximal creep compliance calculated from creep-recovery curves. * indicates statistical significance (*p* < 0.05) compared to untreated control. Infection and Drug Resistance 2022:15 851-871. Originally published by and used with permission from Dove Medical Press Ltd. [121].

**Table 1 ijms-24-02104-t001:** Antimicrobial activity of monometallic NPs. Abbreviations: Au, gold; Ag, silver; Fe_3_O_4_, iron (II, III) oxide; Pd, palladium; TiO_2_, titanium dioxide; ZnO, zinc oxide; Se, selenium.

NPs	Size (nm)	Synthesis	Pathogens	Mechanism of Action	Reference
Au	~37–53 (AuR), ~55–65 (AuP), ~243 (AuR)	Chemical reduction	*Staphylococcus aureus*, *Staphylococcus epidermidis*, *Klebsiella pneumoniae*, *Klebsiella oxytoca*, *Pseudomonas aeruginosa*	Induction of oxidative stress, increase of cellular membranes permeability, cell membrane depolarization, protein leakage from the bacteria, destruction of pathogen’s membranes	[117]
Au	~37–53	Chemical reduction	*Candida glabrata*, *Candida krusei*, *Candida albicans*, *Aspergillus fumigatus*, *Aspergillus flavus*, *Cladosporium herbarum*, *Fusarium oxysporum*	disrupt the outer fungal membrane and increased permeability of *Candida* cells, the release of proteins from damaged *Candida* cells, ROS generation	[120]
Ag	~11–18	Green sythesis (plant)	*Staphylococcus aureus*	Disruption of membrane integrity and permeability, membrane depolarization, decline in efflux pump activity	[155]
Ag	~15–37	Green sythesis (plant)	*Ralstonia solanacearum*, *Xanthomonas axonopodis pv. punicae*	Generation of ROS, disruption of replication and DNA damage	[156]
Fe_3_O_4_	~9	Co-precipitate methods	*Escherichia coli*, *Pseudomonas aeruginosa*, *Staphylococcus aureus*, *Candida albicans*	Nonspecific interaction with membrane compounds, disorganization of lipid packing in the membrane of the microorganism, disruption of transport across the membrane, disruption of cell division	[157]
Pd	13–18	Biosynthesis (plant)	*Staphylococcus auerus*, *Streptococcus pyogenes*, *Bacillus subtilis*, *Enterobacter aerogenes*, *Klebsiella pneumoniae*, *Proteus vulgaris*	The cell membrane destruction and cell apoptosis	[158]
TiO_2_	2–23	Laser ablation in liquid	*Escherichia coli*, *Pseudomonas aeruginosa*, *Proteus vulgaris*, *Staphylococcus aureus*	The interaction between NPs and the cell wall of the microorganism, leading to the microorganism oxidation and finally death	[159]
ZnO	15–30	Biosynthesis (plant)	*Staphylococcus aureus*, *Pseudomonas aeruginosa*, *Escherichia coli*, *Aspergillus niger*, *Aspergillus flavus*, *Aspergillus fumigates*	The production of ROS leading to DNA damaging, denaturation of proteins, rupture of enzymes, and depletion in antioxidant glutathione level causing the cell death	[160]
Se	~110	Biosynthesis (cow urine)	*Staphylococcus aureus*, *Escherichia coli*, *Klebsiella pneumoniae; Pseudomonas* spp., *Serratia marcescens*, *Proteus mirabilis*	The generation of ROS and proteins denaturation	[161]
Se	~55	Biosynthesis (plant)	*Bacillus subtilis*, *Escherichia coli*	The cell wall destruction, inhibiting cell wall synthesis or inactivating other cellular processes	[162]

**Table 2 ijms-24-02104-t002:** Antimicrobial activity of metallic NPs in combination with conventional antibiotics/fungicides or compounds other than antimicrobial agents. Abbreviations: Ni, nickel; Cu, Copper; Zn, zinc; AZI, azithromycin; GEN, gentamycin; OXA, oxacillin; CEFO, cefotaxime; NEO, neomycin; AMP, ampicillin; SUL, sulbactam; CEFU, cefuroxime; FOS, fosfomycin; CHL, chloramphenicol; OXY, oxytetracycline; ERY, erythromycin; CEP, cephacothin; CLI, clindamycin; TET, tetracycline; AMO, amoxycillin; CIP, ciprofloxacin; CEFP, cefpodoxime; CEFI, cefixime; KET, ketoconazole; STR, streptomycin, AMB, amphotericin B; FLU, fluconazole; PVA, polyvinyl alcohol; PAH, polyallylamine hydrochloride; H_2_O_2_, hydrogen peroxide; NAC, N-acetyl-cysteine; TOB, tobramycin.

NPs	Size (nm)	Synthesis	Pathogens	Mechanism of Action	Reference
Ag NPs, ZnO NPs in combination with: AZI, GEN, OXA, CEFO, NEO, AMP/SUL, CEFU, FOS, CHL, OXY	15–16 (Ag NPs), 187–188 (ZnO NPs)	biosynthesis (plant)	*Staphylococcus aureus*, *Salmonella enterica* subsp. *Bukuru*, *Escherichia coli*, *Candida albicans*	The electrostatic interaction between positively charged nanoparticles and negatively charged bacterial cell, release of ions, disruption the cellular respiratory chain, inhibition of unwinding of DNA, the ROS generation	[77]
Ag NPs in combination with: ERY, AMP, CHL, CEP, CLI, TET, GEN, AMO, CIP, CEFP, CEFU	~26	chemical reduction	*Staphylococcus aureus*, *Streptococcus mutans*, *Streptococcus oralis*, *Streptococcus gordonii*, *Enterococcus faecalis*, *Escherichia coli*, *Aggregatibacter actinomycetemcomitans*, *Pseudomonas aeruginosa*	The attachment to the bacterial cell membrane and pore formation, interacting with intracellular biomolecules such as DNA, cause inhibition of DNA replication leading to the cell death, disruption the respiratory chain, transport of hydrophilic antibiotics to the cell surface	[166]
Ag NPs, Ni NPs, Cu NPs, Zn NPs in combination with: CEFI	~17−41	chemical reduction, biosynthesis (plants)	*Salmonella typhi*	Change in the membrane permeability, ROS generation, ATP depletion, DNA damage and disruption, interaction with sulfur and phosphorus-containing molecules	[184]
Ag NPs in combination with: KET	11–19	chemical reduction	*Malassezia furfur*	The attachment and anchoring to the surface of the fungus, ROS generation, leading to structural changes and damage, such as permeability and the membrane potential, forming pores causing leakage of various substances, disrupting the activity of respiratory chain enzymes	[185]
Ag NPs in combination with: STR, AMB, FLU	11–15	biogenic synthesis	*Pseudomonas aeruginosa*, *Escherichia coli*, *Klebsiella pneumoniae*, *Bacillus cereus*, *Candida albicans*, *Candida glabrata*	The interaction with the bacterial cell wall, membrane damage, destruction the proton pump, blocking the metabolism and respiration, pores that disrupt the membrane electrical potential, ROS generation, DNA damage, depletion of glutathione, lipid peroxidation, release of Ag^+^ ions, translation inhibition	[186]
Au NPs in combination with: colicin	35–70	green method	*Klebsiella pneumoniae*	No data	[187]
PVA@Ag NPs and PAH@Ag NPs in combination with: H_2_O_2_	11–15 (PAH@Ag NPs) 17–26 (PVA@Ag NPs)	chemical reduction	*Escherichia coli*, *Staphylococcus aureus*	The electrostatic interaction with the bacterial cell wall, ROS generation, disruption of bacterial cell membrane	[188]
Ag NPs in combination with: ebselen	2–24	biosynthesis (microbes and plant)	*Escherichia coli*, *Staphylococcus aureus*	The ROS generation, interruption of bacterial antioxidant system	[189]
ZnO NPs in combination with guava leaf extract	15–30	chemical reduction	*Escherichia coli*	No data	[190]
Au NPs in combination with: NAC, TOB	~44 (spherical NPs), ~60 (rod-shaped NPs, ~144 (star-shaped NPs)	chemical reduction	*Pseudomonas aeruginosa*	Induction of oxidative stress leading to subsequent permeabilization of microbial membranes and leakage of intracellular contents	[121]

**Table 3 ijms-24-02104-t003:** Antimicrobial activity of multimetallic NPs. Abbreviations: Pt, platinum; CdO, cadmium (II) oxide; NiO, nickel(II) oxide; Fe_2_O_3_, iron(III) oxide; CuO, copper(II) oxide.

NPs	Size (nm)	Synthesis	Pathogens	Mechanism of Action	Reference
Au-Pt	1–3	chemical reduction (Turkevich method)	*Candida albicans; Pseudomonas aeruginosa; Staphylococcus aureus*	ROS generation	[208]
Ag-Au	3–40	chemical reduction	*Xanthomonas oryzae; Magnaporthe grisea*	The damage the bacterial cell wall and release of metal ions	[209]
Ag-Cu and Cu-Zn	80 (Ag-Cu), 100 (Cu-Zn)	biosynthesis (plant)	*Alcaligenes faecalis*, *Staphylococcus aureus*, *Citrobacter freundii*, *Klebsiella pneumoniae*, *Clostridium perfringens*	ROS generation	[210]
Au-Pt-Ag	35–40	biosynthesis (plant)	*Staphylococcus aureus*, *Enterococcus faecalis*, *Escherichia coli*, *Candida albicans*	Interaction with cell membranes (membrane disruption, changes in its permeability); ROS generation, inactivation of some enzymes; destruction of microbial DNA/RNA; lysis of microbial cells	[211]
CdO-NiO-Fe_2_O_3_	~7–28	self-combustion method	*Escherichia coli*, *Pseudomonas aeruginosa*, *Moraxella catarrhalis*, *Staphylococcus aureus*	The ROS generation, release of heavy-metal ions, interaction of nanoparticles with the cell wall of bacteria	[212]
CuO-NiO-ZnO	5–9	co-precipitation method	*Escherichia coli*, *Staphylococcus aureus*	The interaction between nanoparticles and bacterial cell wall. The bacterial cells are ruptured and cracked with the release of intracellular components	[213]
Ag-ZnO-TiO	80–140	sol-gel method	*Escherichia coli*	The inhibition the enzymes for ATP hydrolysis and expression of ribosomal proteins by hindering DNA replication of bacteria and ROS generation	[214]

**Table 4 ijms-24-02104-t004:** Antimicrobial activity of metallic nanoparticles functionalized with antibiotics/fungicides or compounds other than antimicrobial agents. Abbreviations: STR, streptomycin; AMP, ampicillin; Au NFs, gold nanoflowers; CIP, ciprofloxacin; IMI, imipenem; GNRs, gold nanorods; FLU, fluconazole; CAS, caspofungin; Fe_2_O_3_, iron(III) oxide; AMB, amphotericin B; NYS, nystatin; ZrO_2_, zirconium dioxide; GA, glutamic acid; CuO, copper(II) oxide; GLYMO, (3-glycidyloxypropyl)trimethoxysilane; 4-HPBA, 4-hydroxyphenylboronic acid; CH, chitosan; PVP, polyvinylpyrrolidone; MPA, 2-mercaptopropanoic acid; CHX, chlorhexidine; Fe_3_O_4_ NPs@NH_2_, aminosilane-functionalized iron(II,III) oxide nanoparticles.

NPs	Size (nm)	Synthesis	Pathogens	Mechanism of Action	Reference
Ag NPs@STR	~31–119	green synthesis	*Escherichia coli*, *Staphylococcus aureus*	The interaction between NPs and microorganism’s cell wall	[218]
Au NPs@AMP	25–50	chemical reduction	*Escherichia coli*, *Bacillus subtilis*, *Staphylococcus aureus*, *Flavobacterium devorans*	The interaction between positively charged nanoparticles and negatively charged bacterial cell, disruption of bacterial membrane integrity, enhanced entry of antibiotic, inhibition of the bacterial proliferation	[236]
Au NFs@ CIP	No data	adsorption method	*Bacillus subtilis*, *Staphylococcus aureus*, *Escherichia coli*, *Pseudomonas aeruginosa*	The interaction between nanoparticles and bacterial cell wall, release of the carried drugs	[223]
Ag NPs@IMI	~63–65	co-reduction method	*Pseudomonas* spp.	The assembly on the bacterial surface, reduction of the expression of Verona imipenemase (VIM) and Imipenemase (IMP) genes involved in resistance, changes in morphology: chromatin condensation and fragmented nuclei	[246]
GNRs @FLU	72–75	chemical reduction	*Candida albicans*	Conjugating fluconazole with AuR NPs enhanced the delivery efficiency of fluconazole to the cell wall of the fungal cells and accelerated their cellular uptake	[247]
Au NPs@CAS	30–50	chemical reduction	*Candida* spp.	The membrane damage as well as cell wall and cell death	[248]
Fe_2_O_3_ NPs@AMB, Fe_2_O_3_ NPs@NYS	12–16 nm (Fe_2_O_3_ NPs@amphotericin B), 13–17 nm (Fe_2_O_3_ NPs@nystatin)	co-precipitation (Massart method)	*Candida* spp.	Membrane disruption, induction of oxidative stress	[239]
ZrO_2_ NPs@GA	~2.5	solvothermal method	*Rhodotorula mucilaginosa*, *Rothia dentocariosa*, *Streptococcus mitis*, and *Streptococcus mutans*	The interaction between nanoparticles and pathogen cell wall or cellular constituents	[249]
CuO NPs@ GLYMO/4-HPBA	117–125	precipitation method	*Escherichia coli*, *Rhodococcus rhodochrous*	The ROS generation, leading to peroxidation of lipids from the bacterial cell membrane, interaction ROS with the cell organelles, electrostatic interaction between nanoparticles and bacterial cell wall, release of free Cu^2+^ ions	[250]
Ag NPs@CH	13–42	green synthesis	*Staphylococcus aureus*, *Pseudomonas aeruginosa*, *Candida albicans*	The electrostatic interaction between nanoparticles and bacterial cell wall leading to the leakage of the intracellular components, penetration the cell wall of the fungus and interaction with sulfur-containing membrane proteins and phosphorus-containing DNA nitrogenous bases	[251]
AgNPs@PVP	6–10	chemical reduction	*Acinetobacter baumannii*	The ROS generation, changing the expression level of proteins	[252]
Au NPs@ MPA-cationic dipeptide	10–14	chemical reduction (Turkevich method)	*Escherichia coli*, *Staphylococcus aureus*, *Candida albicans*, *Candida glabrata*	The disruption of the cell wall	[253]
Fe_2_O_3_ NPs@CHX	10–14	co-precipitation (Massart method)	*Staphylococcus aureus*, *Enterococcus faecalis*, *Escherichia coli*, *Pseudomonas aeruginosa*, *Candida* spp.	Depolarization of mitochondria, induction of oxidative stress and oxidation of pathogen structures	[147]
Fe_3_O_4_ NPs @Au, Fe_3_O_4_ NPs @NH_2_	10 (Fe_3_O_4_ @Au), 13 (Fe_3_O_4_ @NH_2_)	co-precipitation (Massart method)	*Pseudomonas aeruginosa*	Attach to the bacterial membrane and loss of its integrity, electrostatic interactions with the bacterial cell wall resulting in damage to the cell wall, increased membrane permeability, perforation of the plasma membrane, disruption of cell metabolism	[134]

**Table 5 ijms-24-02104-t005:** The influence of NPs such as Ag, TiO_2_ and ZnO on the intestinal microbiota. Abbreviations: Ag, silver; TiO_2_, titanium dioxide; ZnO, zinc oxide; F/B, Firmicutes/Bacteroidetes: ↑, increased bacteria; ↓ decreased bacteria.

NPs	Effect		Reference
Ag	↓ Firmicutes ↑ Bacteroidetes ↓ *Lactobacillus* ↑ *Bifidobacterium*		[291,294]
	↑ F/B ratio with dose ↑ *Coprococcus* ↑ *Lactobacillus* ↑ *Blautia* ↓ *Bacteroides* ↓ *Mucispirillum*		[291,295]
	↓ F/B ratio ↑ Alistipes ↑ *Bacteroides* ↑ *Prevotella* ↓ *Lactobacillus*		[291,296]
	↓ Bacteroidetes i.e., *Bacteroides ovatus* ↓ *Eubacterium rectale* ↓ *Faecalibacterium prausnitzii* ↓ *Roseburia faecalis* and *Roseburia intestinalis* ↓ *Ruminococcus torques* ↑ *Escherichia col* *i*↑ *Raoultella* (sp.)		[290,297]
TiO_2_	↓ *Bacteroides ovatus* ↑ *Clostridium cocleatum*		[291,298]
	No considerable effect on gut microbiota. Microbial composition and GIT histology remained unchanged.		[291,296]
	↑ *Lactobacillus reuteri* ↓ *Romboutsia*		[291,299]
	↓ *Bacteroides ovatus* ↑ *Acidaminococcus intestini* ↑ *Clostridium cocleatum* ↑ *Eubacterium rectale* and *Eubacterium ventriosum*		[290,298]
ZnO	↑ Streptococcus ↓ *Lactobacillus*	Ileum	[290,291,300]
	↑ *Lactobacillus* ↓ *Oscillospira* ↓ *Prevotella*	Colon	
	↓ Firmicutes ↓ *Lactobacillus* ↑ Bacteroidetes ↑ *Fusobacteria* ↑ Bacilli		[291,301]

## Data Availability

No new data were created or analyzed in this study. Data sharing is not applicable to this article.

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
