# Peer review of "Metallic Nanosystems in the Development of Antimicrobial Strategies with High Antimicrobial Activity and High Biocompatibility"

_ijms, 2023, doi:10.3390/ijms24032104_

Round 1
Reviewer 1 Report
The manuscript entitled “Metallic nanosystems in the development of antimicrobial strategies with high antimicrobial activity and high biocompatibility”, submitted for evaluation to IJMS, presents the summary of metallic nanoparticles and nanosystems based on metallic nanoparticles containing antimicrobial agents attached to their surface used for replacement of support of antimicrobial drugs in biomedical applications. In the context of the increase in the microbial resistance and emergence of multi-drug resistant bacterial strains, I consider the topic of the work to be important and timely.
In general, the work clear and written in the correct language. The structure of the review is clear. The selected literature reports were divided into several 5 sections concerning the overall list of nanostructures used in biomedicine as well as action mechanisms of metallic NPs, synergistic effects of metallic nanoparticles, biocompatibility of metallic nanoparticles. This way of presentation makes the review of the presented reports simple and clear for the reader.
I only some minor errors. My comments and questions concerning the submitted article are listed below:
COMMENTS TO AUTHORS
1. Some language mistakes appear, e.g. in: “The difference in antimicrobial activity of these Ag NPs were explain by the release rate of Ag ions from the surface.” (explain should be replaced by explained). Similarly, the sentence “Despite the presence of beta-lactamase and carbapenemase resistance genes in Acinetobacter baumannii, the combination of AgNPs with imipenem is an effective against carbapenem-resistant strains, showing potent antimicrobial activity [236].” should be corrected.
2. Pages 6-7: Two fragments describe one of the aspects of action of gold NPs (disruption of cell membrane): “Gold nanoparticles, like silver nanoparticles, disrupt the integrity and structure of the cell membrane, causing leakage of intracellular components [113-115].” and “One hypothesis for gold nanoparticles antimicrobial mechanism of action consider the apoptosis-like cell death, where gold nanoparticles cause depolarization of the bacterial cell membrane and a continuous increase in the concentration of calcium ions in the cytoplasm, induction of DNA fragmentation, resulting in apoptosis-like death (overexpression of caspase-subunit proteins was observed as well) [119].”. Please reedit this fragment of manuscript for better clarity.
3. Page 10: Is the following fragment “In addition, metallic nanoparticles can also release metal ions which can penetrate the cell and thus interfere with biological processes, among others by generating ROS. As a result of disruption of redox homeostasis, glutathione is oxidized, thus inhibiting the antioxidant defense mechanism of bacteria against ROS. Then, the released metal ions may interact with cellular structures by forming coordination bonds with nitrogen, oxygen or sulfur atoms, which are incorporated into intracellular structures (e.g. proteins or DNA) and disrupting cellular functions [62]. Due to non-specific binding between metal ions and bio-molecules, metallic nanoparticles generally exhibit broad-spectrum of antibacterial activity [75].” necessary in this section? The release of metal ions and the consequences of this phenomenon to bacterial cells function was already mentioned in the manuscript…
4. Page 10: I do not follow the sense of the sentence: “The results indicate that the increase in solubility obtained by increasing the concentration of the plant extract used for the synthesis in the half maximal inhibitory concentration values were found to decrease (from 4% to 21%) against Escherichia coli, Pseudomonas aeruginosa, Staphylococcus aureus and Micrococcus luteus strains [136].”
5. Page 15: “Bimetallic nanoparticles Ag-Au NPs synthesized by the core-shell method [190] showed synergistic antimicrobial activity of bimetallic nanoparticles conjugated with doxycycline against Pseudomonas aeruginosa and Escherichia coli, where the combinatorial effect led to higher drug binding affinity and enhanced antimicrobial efficacy.” This fragment should be moved to the section 2.3. Multimetallic nanoparticles.
6. What is missing in this review is the section reporting the NPS-resistance development in bacteria (which occurs rarely but was reported in several articles) and their effect on the host microbiome. I strongly encourage to include this additional section in the revised version of the manuscript.
Reviewer 2 Report
The article ijms-2141250 entitled "Metallic nanosystems in the development of antimicrobial strategies with high antimicrobial activity and high biocompatibility" has been submitted by Karol Skłodowski, Sylwia Joanna, Chmielewska, Ewelina Piktel, Przemysław Wolak, Tomasz Wollny, and Robert Bucki for publication in the International Journal of Molecular Science.
This article is a review of literature on metallic nanomaterials used for biomedical applications. In particular, their antimicrobial activity is thoroughly described.
After a detailed introduction of the topic, the authors describe the synergistic effects of metallic nanoparticles, the properties of monometallic nanoparticles, their combination with antibiotics, the properties of multimetallic nanoparticles, and their ability to carry molecules with antimicrobial activity.
At last, the biocompatibility of the metallic nanoparticles is reviewed.
In my opinion, this review article is really good. It contains a lot of relevant information supported by 282 references that will be of great interest for the readership of the journal.
I recommend this review article to be accepted and published in the International Journal of Molecular Sciences.
